# Severe Drought Monitoring by Remote Sensing Methods and Its Impact on Wetlands Birds Assemblages in Nuntași and Tuzla Lakes (Danube Delta Biosphere Reserve)

**Vasile Jitariu** [1] **, Alexandru Dorosencu** [2] **, Pavel Ichim** [1] **and Constantin Ion** [3,*]

[1] Faculty of Geography and Geology, Alexandru Ioan Cuza University, 700506 Iasi, Romania; vasile.jitariu@uaic.ro (V.J.); pavel.ichim@uaic.ro (P.I.)

[2] Danube Delta National Institute for Research and Development, 165 Babadag Street, 820112 Tulcea, Romania; alexandru.dorosencu@ddni.ro

[3] Faculty of Biology, Alexandru Ioan Cuza University, 700506 Iasi, Romania

[*] Correspondence: constantin.ion@uaic.ro

**Abstract:** The present paper aims to highlight the impact of the partial or total drying of the Nuntași and Tuzla lakes (from the Danube Delta Biosphere Reserve) as a result of intense drought phenomena on groups of waterfowl that are encountered in this region. Our analysis combined satellite remote sensing techniques with bird observations that were made monthly during the analyzed period, corroborated with the meteorological context of the time interval that was taken into account. The results of the satellite image processing show a partial drying in 2013 and a total drying in 2020 of the Nuntași and Tuzla lakes, which were caused by both natural factors (drought) and anthropogenic factors (inadequate management of the area—e.g.,: communication channels with surrounding lakes are clogged). These situations have led to repercussions for groups of birds, which behave differently depending on their ecology. Pelicans and swans are the most affected birds, they leave the area in the absence of water, whereas gulls and terns are not affected by the decrease in the water surface, they even increase their numbers in such conditions. Our study also shows that from 2010 to 2020 the largest numbers of birds (total numbers of birds), with the exception of pelicans, were recorded in 2013 and 2020, more precisely in the years when the water surface decreased considerably. Another important feature of this paper involves highlighting how fragile an ecosystem can be in the context of climate change, but also how important it is to involve human society in maintaining the adequate conditions for an ecosystem that is part of one of the most important biodiversity hotspots on the planet, the Danube Delta Biosphere Reserve.

**Keywords:** remote sensing; drought impact; aquatic birds; Danube Delta Biosphere Reserve

## 1. Introduction

Droughts are one of the most frequent meteorological disasters that occur on a large scale, with negative implications in several fields of human society [1]. Whether we are talking about meteorological drought, agricultural drought or hydrological drought [2], this phenomenon makes its repercussions manifest on multiple levels, but, nevertheless, it is a phenomenon that is difficult to anticipate [3] and complex when it comes to analysis, which is why a number of indicators have been designed in order to highlight its intensity [4,5].

Drought can be considered the most complex climatic phenomenon [6] because it triggers several factors, such as: atmospheric precipitations, air temperature, humidity, soil water reserves evapotranspiration, wind speed etc.; these being the main climatic parameters that define the state of dry weather. Drought spell occurrence is directly influenced by the characteristics of the active surface, soil characteristics, plants' physiological peculiarities and the consequences of the anthropogenic influence on the environment [6]. Drought is normally classified into one of three types: meteorological drought (characterized by a

lack of precipitation), hydrologic drought (a lack of water supply, declining river flow and groundwater supply) and agricultural drought (crop water deficit) [2]. In Romania, the highest frequencies of drought phenomena are specific to the eastern, northeastern and southeastern regions, including the Danube Delta Biosphere Reserve [7,8].

This paper will analyze the influence of hydrological drought on the variation of the water surface (WS) of the Nuntași and Tuzla Lakes and the impact of its drying in two representative periods that were captured on satellite images. Hydrologic drought is defined as a period during which stream flows are inadequate to supply the established uses under a given water management system [9].

The drought phenomenon, depending on the region in which it takes place, can produce changes/anomalies in biodiversity. For example, in arid and semi-arid regions, biodiversity responds very quickly to precipitation [10] and in the case of the Sonoran Desert, Mc Creedy and van Riper, 2014 [11] observed a delay in nest initiation for 15 species of birds in 2006–2007. Magoulick and Kobza (2003) [12] mentioned that, for freshwater areas, severe drought phenomena can reduce the habitat area and can lead to changes in fish behavior (predator and competition relations) because drought leads to shifts in refuge spacing. Batanero et al. (2017) [13] have argued that intense droughts lead to a more pronounced increase in wetland eutrophication, especially where inclusive large colonies of birds can be found. In order to understand drought at different temporal scales, bird abundance and distribution may better reveal which drought factors impact on birds and therefore improve our understanding of how climate change impacts species and the landscapes that they inhabit [14]. Little is known about the responses of birds with different functional and behavioral characteristics to drought, or how these responses vary geographically across large areas [15]. Moreover, microclimate variables are the key factors that influence the distribution, diversity and density of the wetland bird species [16].

Waterfowls' abundance and species richness depends especially on the size, shape and depth of a body water. Moreover, the presence of water could assure the development of the birds' food and create places for birds to roost [17].

Satellite remote sensing is an alternative through which spatio-temporal observations can be made regarding changes in the earth's surface and, by this method, depending on the state of the vegetation [18] or the fluctuation of the WS [19], some conclusions can be drawn regarding the severity of the drying occurrence and the variation of the WS. Monitoring microclimate variables and birds is possible in order to understand the fluctuation of bird assemblages in the context of climatic changes.

Although the literature dealing with aspects of remote sensing, climatology and ecology is still scarce, with this paper we want to outline the importance of multidisciplinarity in science through which, in addition to the classical climate–biodiversity analysis, we can track changes in habitats through satellite remote sensing and underline the impact on aquatic birds.

The broader impact of the study is that the presence of birds could be correlated with remote sensing analysis and the weather parameters' fluctuation. The impact of climate change over the distribution of the precipitation amount in Romania was analyzed by Croitoru et al., 2018 [20]. The climate scenarios of the Representative Concentration Pathways (RCPs) 4.5 were used. Using those climate scenarios, it was observed that, in the future, the Dobruja region will record a decrease in its precipitation amount of approximately 90% as compared to the historical climate data that were recorded during dry spell events, which will intensify drought periods [20]. The fate of birds will be disastrous if the drought is not stopped through adequate management measures regarding the hidrorological system, bykeeping the channels unclogged.

This study aims to highlight the impact of the Nuntași and Tuzla Lakes' WS area variations on their aquatic bird groups in the context of the current climate trends through remote sensing methods combined with data from avifauna field monitoring. Our study focuses on the years 2013 and 2020 because the most intense modifications of the WS were

observed from the field information at that time and this was also observed by Serban et al., (2022) [21].

## 2. Study Area

The Nuntași–Tuzla complex is a natural reserve that is included in the Danube Delta Biosphere Reserve and Natura 2000 network of protected areas (SPA and SCI) and it represents one of the main hotspots for birds in Eastern Europe [22] (Figure 1).

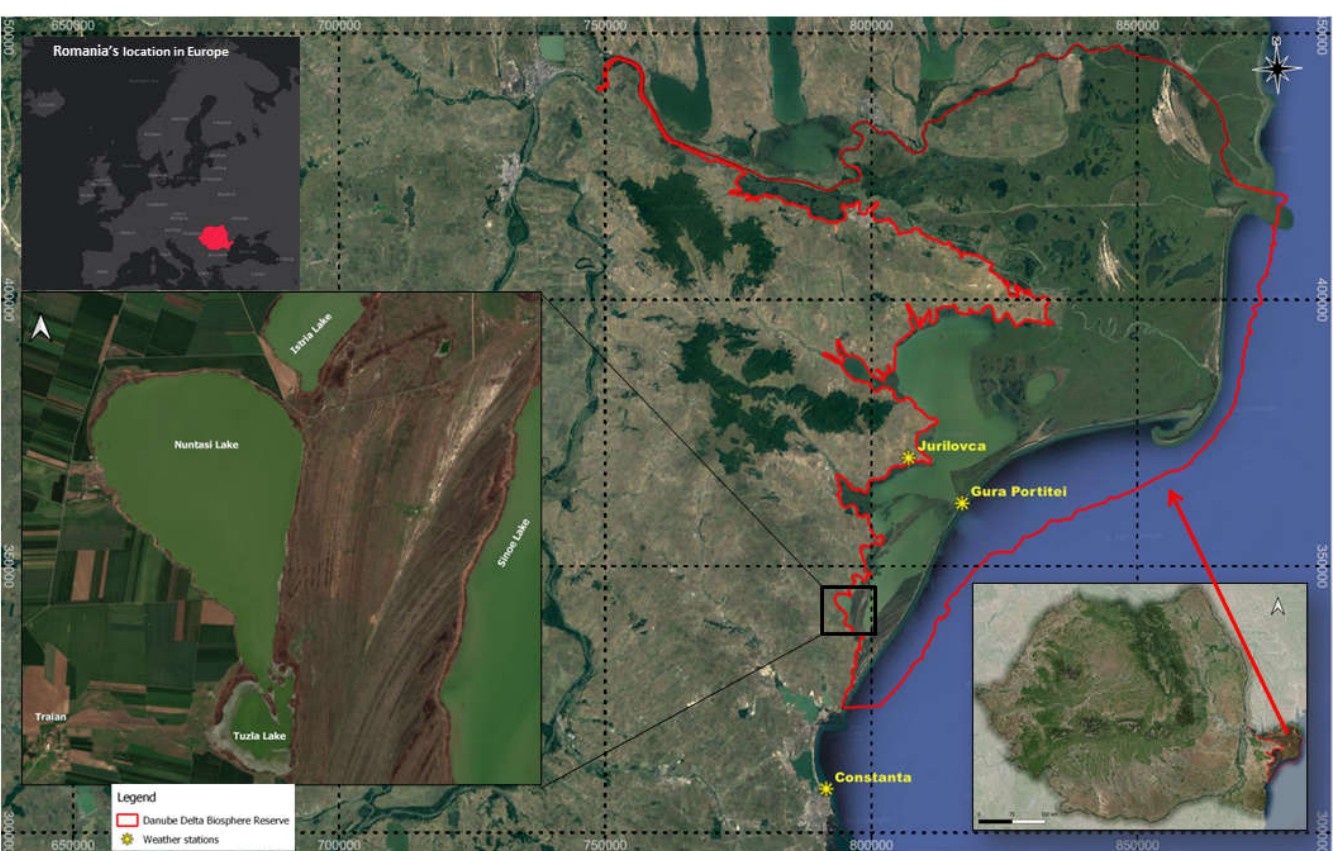

**Figure 1.** Location of Nuntași–Tuzla complex inside the Danube Delta Biosphere Reserve.

The medium-sized shallow lakes Nuntași–Tuzla are situated between 8–11 km west of the Black Sea shoreline [22–24]. Among the main characteristics concerning these lakes' morphometry, they have a length of about 2 km, a maximum width of 3 km and a depth that varies between 2.15 and 6.15 m [25]. There are marginal lakes of the Razim–Sinoie lagoon complex that resulted when the natural levees had appeared along the former sea's bay. The lakes have been undergoing a slow process of desalination in the last half a century. Now the water is brackish and waves stir up the sediments that cause the lakes to be turbid when it is windy. It is known that most brackish lakes are turbid and that this state may be due to the absence of phytoplankton grazers (*Daphnia* sp.) [26]. Being turbid, the only places that are suitable for submerged vegetation growth are some marginal shallow and covered sites. In some years, floating algal beds cover large parts of the lakes as a response to the high availability of nutrients in the water column. Common reed (*Phragmites australis*) dominates the shore vegetation and forms several islands in the southern half of the lake system. Two rivers (the Nuntași and Săcele) supply the north-western and south-western extremities, respectively, of the Nuntași–Tuzla lakes complex with fresh water that is collected from precipitation and two man-made channels that interconnect it with Lake Istria in the north and Lake Sinoie in the south. The northern channel provides the most significant water supply for the complex.

The lakes are surrounded by extensive cereal and oilseed rape fields to the north, west and south (Figure 1), these are excellent autumn and winter feeding places for the goose and swan populations. The WS represents a safe night-roost and an important drinking source for the waterbird populations of the area [27].

In the spring and autumn, the complex is heavily used by migrating waterbirds. The lakes are an important feeding and staging area for migrating waterbirds in the so-called Via Pontica flyway and one of the key roosting sites for geese, swans, pelicans and shelducks. The shallow water and muddy fields of the lakes attract waders and gulls in great numbers.

The WS varies due to both natural and anthropic factors [24]. In 2013 and 2020, the lake dried (partially and totally, respectively) because of climatic factors and management measures.

The choice of the study area was motivated by the drying phenomenon in 2020, which, due to its visual impact, has also been the subject of a considerable number of press articles and TV reports. Given the fact that this complex of lakes is located in the Danube Delta Biosphere Reserve, a biodiversity hotspot, we considered it necessary to carry out a study in order to examine the implications of this phenomenon on the avifauna of the region.

## 3. Methodology

### 3.1. Bird Census

Waterbird species surveys have been carried out once per month using four points that offer the optimal visibility over the lakes. Individual count units have been used in order to reduce bias. The counts have been performed regularly in the morning and evening, including the roosting sites.

The data were collected in such a way as to avoid the double counting of the same individuals from two neighboring points and the resulting numbers were summed in order to obtain the totals per location. The birds were identified with binoculars and field scopes [27,28].

We focused, in this analysis, on aquatic bird assemblages, such as: geese, swans, dabbling ducks (they live in shallow waters and feed by tipping up rather than diving), shelducks, gulls, terns, waders and pelicans. We took into consideration only these birds, because they represent a constant presence on the Nuntași and Tuzla Lakes, which are used mainly for feeding (by gulls, terns, waders, the ruddy shelduck), roosting (by geese and swans) or both (by ducks and the common shelduck).

### 3.2. Climate Data

To understand the climate characteristics of the region we used the ROCADA gridded daily climatic dataset over Romania (1961–2013) that was developed by Dumitrescu & Bârsan in 2015 [29]. For the latest period (after 2014) we used daily observed meteorological data that were obtained from the National Meteorological Administration of Romania (NMA RO) from SYNOP messages that were issued by official weather stations. The daily data that were analyzed were obtained from 3 weather stations (Gura Portiței, Jurilovca and Constanța). The daily data that we used cover the main weather elements were: the mean daily air temperature, minimum temperature, maximum temperature and precipitation (daily rain amount and number of days with precipitation) in 2013 and 2020.

### 3.3. Satellite Data

Satellite imagery is an excellent way to track various phenomena, both natural and artificial, but it must be admitted that one of the main disadvantages of this research method is the availability of the scenes, which is why, in some situations, the involvement of other methods is necessary (field data, meteorological context, etc.). The analysis of a phenomenon by satellite remote sensing methods can be hampered either by an overloaded atmosphere, clouds, or too much time between scenes.

From the perspective of satellite images, depending on the analyzed period and the availability of materials, we chose two sets of images. Landsat-5 satellite images were obtained through Google Earth Engine (GEE) for 2013 and, due to the fact that the study

area is not very large, we downloaded the scenes without clouds, without using a mosaic of images from several days. Although in 2013 we could use other satellites with better resolution, we continued with Landsat-5 because we identified cloudless sequences for a longer period of time. The Landsat-5 TM sensor contains seven bands: blue, green, red, NIR, TIR, and two TIR bands (Table 1). For the year 2020 we used Sentinel-2A/2B satellite images, images with a very good resolution (10 m, 20 m), 13 spectral bands and with a high frequency of revisitation (Table 1).

**Table 1.** Specifications of Landsat-5 and Sentinel-2.

| Satellite | Sensor | Year | Day/Month | Resolution (m) | Wavelength (μm) |
|---|---|---|---|---|---|
| Landsat-5 | TM | 2013 | 2/May 6/August 22/August 7/September | 30 | Band 1: 0.45–0.52 Band 2: 0.52–0.60 Band 3: 0.63–0.69 Band 4: 0.76–0.90 Band 5: 1.55–1.75 Band 6: 10.40–12.50 Band 7: 2.08–2.35 |
| Sentinel-2 | 2A/2B | 2020 | 26/June 11/July 31/July 25/August 9/September 19/September 24/October 23/November | 60 10 10 10 20 20 20 10 20 60 60 20 20 | Band 1: 0.443 Band 2: 0.490 Band 3: 0.560 Band 4: 0.665 Band 5: 0.705 Band 6: 0.740 Band 7: 0.783 Band 8: 0.842 Band 8A: 0.865 Band 9: 0.945 Band 10: 1.375 Band 11: 1.610 Band 12: 2.190 |

*3.4. Data Analysis*

In order to observe the fluctuations of the water level of Lake Nuntași, we considered the use of two indices. The normalized difference water index (NDWI) is one of the most frequently used indicators to identify open water features being developed in this direction by McFeeters (1996) [30], but later being used in the classification of wetlands [31], the mapping of tidal flats (Murray, 2012) [32] and in the mapping of the spatiotemporal changes of some lakes [33].

The normalized difference vegetation index is a well-known index that has a high degree of applicability, from the identification of land use/cover [34,35] and the monitoring of vegetation health [36,37], to the monitoring of drought phenomena [38,39] and afforestation and deforestation phenomena [40,41]. This index is also used in monitoring wetlands or the water levels of some lakes [42–44].

We used these two indicators in parallel to obtain information on the variation of the water body's surface and to compare the results, so that the final data are as close as possible to the real situation, especially since both indicators reveal extremely useful information about the presence/absence of water.

The *NDWI* can be obtained from a division that is based on the *NIR* band and green band as in Equation (1) as shown below [30] from which we will distinguish the WS as the areas where we identify values that exceed the threshold of "0" [33,45].

$$NDWI = \frac{(Green - NIR)}{Green + NIR} \tag{1}$$

The *NDVI*, shown in the following Equation (2), is explained by the fact that the reflectance of the vegetation in the *NIR* band is higher than that which is found in the *Red*

band and, in this way, the vegetation is easily observed. Water bodies are associated with *NDVI* values that are less than 0 [46].

$$NDVI = \frac{NIR - Red}{NIR + Red} \tag{2}$$

After obtaining the aforementioned indices, we reclassified the rasters and extracted only the "water" values (mentioned above), resulting in a "mask" of the water surface for each scene that we used.

Before modeling the birds, meteorological data and the WS, we plotted the total number of birds in those years with severe droughts, 2013 and 2020. Consequently, we analyzed the total number of birds for the 2010–2020 period, in order to see the differences between those years when the lakes had water and those with a lack of water. The differences were checked using a Kruskal–Wallis test as is mentioned by Marusteri and Bacarea, 2009 [47]. The statistical analyses were computed using Poisson regression, which is a generalized linear model that has been used in order to elucidate the relationships between biological assemblages of species and their environment, as suggested by Van Strien et al., 2004 [48]. Given that the histograms regarding the number of birds that were analyzed per month indicated a deviation from the normal distribution and that the abundance values for each group of birds are inclined to the right in the Poisson distribution, we used Poisson regression [49,50]. We computed, in XLSTAT software, the biological assemblages (the average number of each group) which were represented by a series of aquatic birds as dependent variables. The explanatory variables consisted of the meteorological data per each month and year, such as the frequency of precipitation, average precipitation, temperature mean, minimum temperature, maximum temperature and the WS in those months for which we have access to satellite images.

## 4. Results

### 4.1. Climatic Background

Although the study area is relatively small, during the analyzed time interval the mean temperature and the amounts of monthly precipitation highlighted strong fluctuations. According to Köppen climate classification, Dobrogea is located in type BSk (semiarid steppe region) as a transition between a Mediterranean climate on its poleward margin and a cooler climate with a mild winter, low precipitation, great variability in the precipitation from year to year, low relative humidity and high evaporation rates [51].

The study area is characterized, from a climatic point of view, by a mean air temperature that can vary between 11.5 and 12 °C (according to the data from the stations that were mentioned above) with the highest temperatures encountered in July and August and the minimum temperatures manifesting in January and February. The average multiannual precipitations are distinguished by quite significant differences in the territory, which fluctuate from 366.7 mm at Gura Portitei station to 424.6 mm at Constanta station. The minimum quantity of rainfalls are typically recorded in February, March and August, while the highest amount of precipitation falls in May and June (Figure 2; Table 2). The low amount of precipitation in August, in the context of high temperatures, leads to an intense process of evaporation and in atypical, particularly dry years this can extend over a longer period. The climatic elements of this study broadly follow the topoclimatic and microclimatic characteristics that formed the basis of the episodes in which Lake Nuntași dried up. A climate description of the region was made by associating the climate elements with the climate region that is specified by the Koppen climate classification.

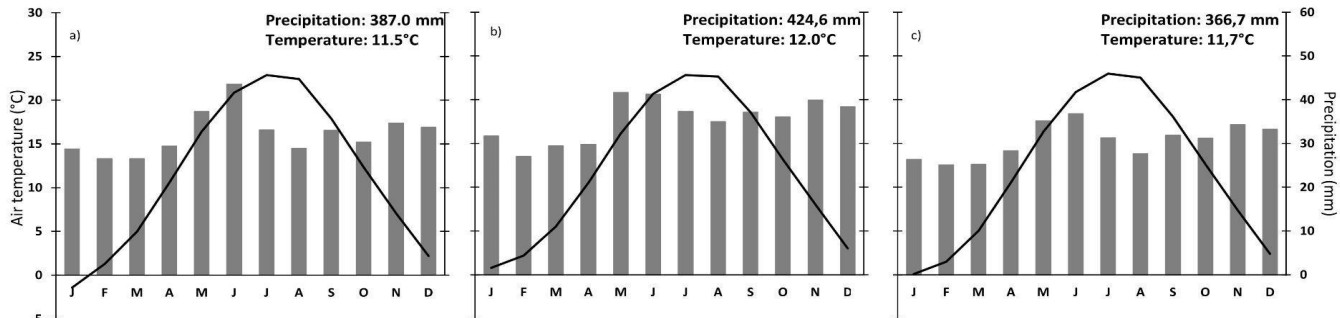

**Figure 2.** The multiannual mean variation of air temperature and precipitations at (**a**) Jurilovca, (**b**) Constanța and (**c**) Gura Portiței weather station from 1961 to 2020.

**Table 2.** Multiannual monthly mean number of days with precipitations over 1 mm and monthly number of days with precipitations over 1 mm in 2013 and 2020 for Jurilovca, Constanța and Gura Portiței weather stations.

|  |  | J | F | M | A | M | J | J | A | S | O | N | D |
|---|---|---|---|---|---|---|---|---|---|---|---|---|---|
| **Mean number** | **Jurilovca** | 4.9 | 5.0 | 4.7 | 5.3 | 7.2 | 6.3 | 5.1 | 4.4 | 4.8 | 4.3 | 5.3 | 5.8 |
|  | **Constanța** | 5.3 | 5.0 | 5.1 | 5.5 | 7.0 | 6.4 | 5.3 | 4.0 | 4.4 | 4.7 | 5.7 | 6.4 |
|  | **Gura Portiței** | 4.9 | 5.1 | 4.8 | 5.4 | 7.3 | 6.7 | 5.2 | 4.4 | 5.0 | 4.6 | 5.6 | 6.2 |
| **2013** | **Jurilovca** | 5 | 5 | 2 | 7 | 7 | 9 | 6 | 4 | 7 | 6 | 4 | 0 |
|  | **Constanța** | 9 | 4 | 4 | 6 | 6 | 8 | 5 | 4 | 6 | 4 | 4 | 2 |
|  | **Gura Portiței** | 6 | 5 | 3 | 7 | 7 | 8 | 6 | 3 | 9 | 6 | 5 | 2 |
| **2020** | **Jurilovca** | 1 | 4 | 1 | 2 | 4 | 6 | 2 | 1 | 1 | 5 | 5 | 10 |
|  | **Constanța** | 1 | 4 | 3 | 1 | 5 | 7 | 3 | 1 | 2 | 4 | 3 | 11 |
|  | **Gura Portiței** | 2 | 5 | 1 | 2 | 5 | 6 | 1 | 0 | 2 | 6 | 5 | 9 |

*4.2. Bird Observation*

As a brief overview of the bird inventory, three species of swans were recorded for the area: the mute swan (*Cygnus olor*), whooper swan (*Cygnus cygnus*) and Bewick's swan (*Cygnus columbianus*). The whooper and Bewick's swans winter in this area and feed on the adjacent arable fields, which include winter cereals (wheat and barley) and oilseed rape. They use the lakes only for roosting and drinking water. The mute swan is present all year around. The number of mute swans constantly varies in the study area depending on the water level and vegetation state (submerged and floating plants). Years with a drastic aquatic surface reductions cause a lack of swans on these lakes.

The red-breasted goose (*Branta ruficollis*), greater white-fronted goose (*Anser albifrons*) and greylag goose (*Anser anser*) are observed in the area in large numbers. Other geese species like the lesser white-fronted goose (*Anser erythropus*) and bean goose (*Anser fabalis*) are recorded in very small numbers alongside large flocks of the greater white-fronted goose. Except for the greylag goose which is also a breeding species in the area, all of the other geese species use the lakes only as roosting and drinking sites during the winter (Figure 3).

Waders use the area mainly for feeding, so their number depends on both the WS and the depth and/or the extent of the mudflats.

In order to observe the link between the different biotic and abiotic factors and the presence of birds, satellite imagery is an excellent way to track various phenomena, both natural and artificial, but it must be admitted that one of the main disadvantages of this research method is the availability of scenes. This is why, in some situations, the involvement of other methods is necessary: including field data, meteorological context, etc.

In 2013 there was a narrowing of the water line, but not a complete drying. We identified a decrease in the water's surface area from about 800 ha (Lake Nuntași and Lake Tuzla) to 300 ha in the May–September period. There was also a greater impact of the drought on Lake Tuzla (in the south), which was the first to dry out and it remained so for the longest period of that year (Figure 4).

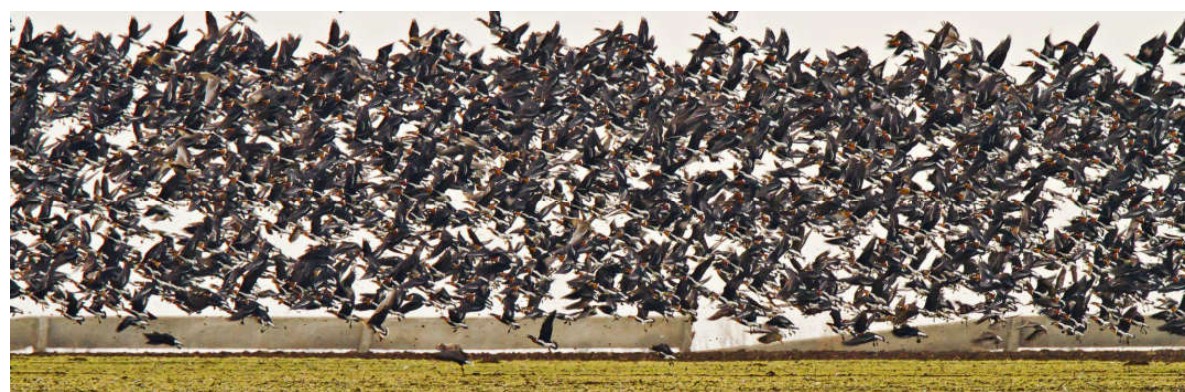

**Figure 3.** Agglomeration of birds (red-breasted goose) which roost in Nuntași–Tuzla lakes.

**Figure 4.** WS in 2013 (Landsat-5) (**above**); Days with representative precipitation amounts at, Jurilovca, Gura Portiței and Constanța weather stations in 2013 (**below**).

Following the reduction of the water's surface in 2013, we identified that the most-affected birds were swans and ruddy shelducks, which were missing in May–August and April–June, respectively (Figure 5). Dabbling ducks were also declining during the droughts. Gulls and terns did not seem to be affected by the restriction of the surface of the water, remaining in large numbers throughout the season. Geese were found in larger numbers during the winter, when specimens come from the north of the continent and are crowded into small areas. Pelicans were found in small numbers in winter because many migrate south, but at the same time their numbers were declining as the WS shrunk, due to their feeding method. Wader birds had low populations due to the droughts. Shelducks seemed to be very versatile, their number decreased both in conditions of drought and when the water level was higher and it was more difficult for them to reach the food at the bottom of the lakes (food that consists of small mollusks, crustaceans and insects) (Figure 4).

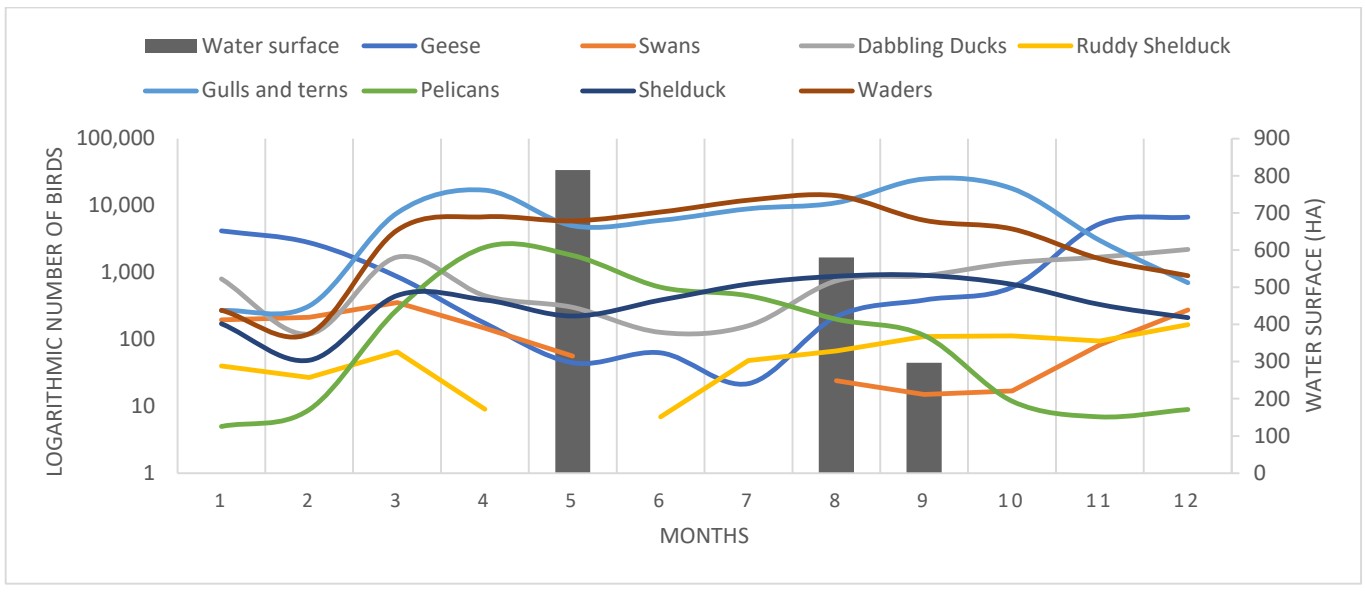

**Figure 5.** Average number of birds per the months in 2013.

For the 2020 drought phenomenon, we used Sentinel-2 satellite images which helped us to track the phenomenon on a more detailed time scale due to the higher frequency of the scenes. The Nuntași–Tuzla lakes have suffered increases and decreases in their WSs. About 80 consecutive days with less than 5% water were identified during August, September and October. In fact, also based on these satellite images, we could observe the dependence of the studied lakes on the climatic factor through the very fast response of the drying up due to the lack of precipitation (Figure 6). The favorable context of this phenomenon was given also by the fact that the winter of 2019–2020 was the warmest winter in the history of meteorological measurements at the Jurilovca weather station and the amount of precipitation was slightly below the average of the entire range of meteorological observations. During that winter, the North Atlantic Oscillation index was in a positive phase, which generated a mild and dry winter in southern and southeastern Europe [52] by the extension and presence of the high atmospheric pressure over the Azores Islands which extended over the Mediterranean basin [53].

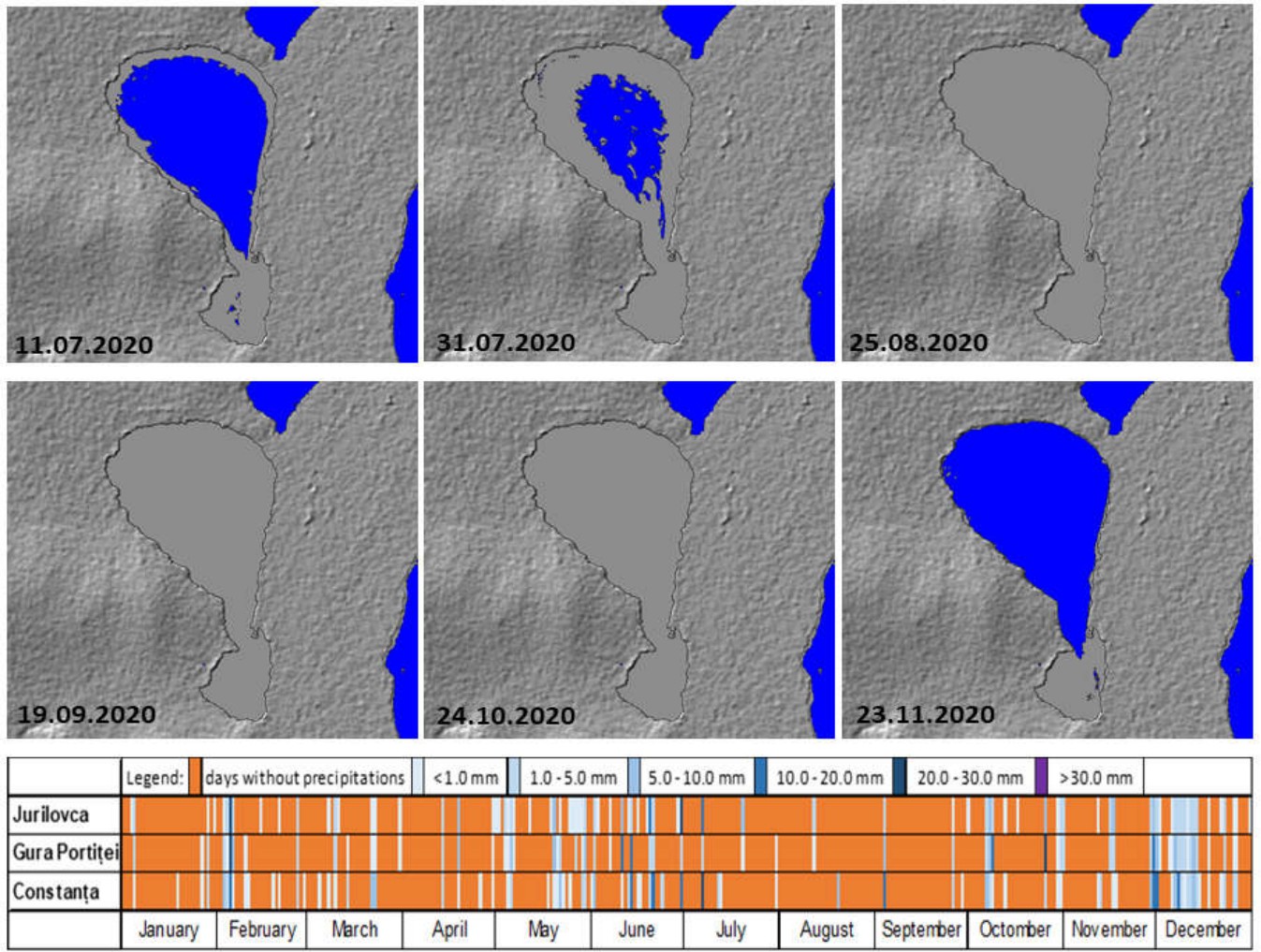

**Figure 6.** WS in 2020 (Sentinel-2) (**above**); Days with representative precipitation amounts at Jurilovca, Gura Portiței and Constanța weather stations in 2020 (**below**).

Given the entire meteorological context, the drought of 2020 manifested itself at a higher intensity on the two lakes and implicitly on the lakes' birds. In contrast to swans and geese, the low lake surface is accessible as a feeding habitat for waders only at a low water level. Dabbling ducks and the common shelduck use the area for roosting and feeding and their numbers are less influenced by the WS and depth, but for duck species there is a gradual shift in the species' presence and abundance according to the season and water depth. The same situation also applies to the ruddy shelduck. Gulls and terns were present throughout the analyzed period and it seems that the presence of these birds is differently affected by water body variations. Their numbers increased considerably when the WS decreased. Pelicans, on the other hand, prefer areas with a certain level of water that facilitates fish presence, so their number is generally low in the years when the lakes may dry out and for the year 2020 they appeared in an extremely small number and for a very short period. The absence of swans was observed for a longer period of time in 2020 than in 2013, as they were not present throughout the warm season (Figure 7). The number of birds was generally higher in 2020 than in 2013, as a result of the total drying of the lakes, but this difference is not statistically significant as is shown below (Figure 7).

In the third decade of November (in the year 2020) there was an almost complete filling of the lakes, a process that was realized after the clearing of the canal that connects Lake Nuntași to Lake Sinoie through Lake Istria. The importance of human intervention in

the well-being of that habitat is undeniable, which is observed after the rapid filling of the lake even in the context of a small amount of precipitation (Figure 8).

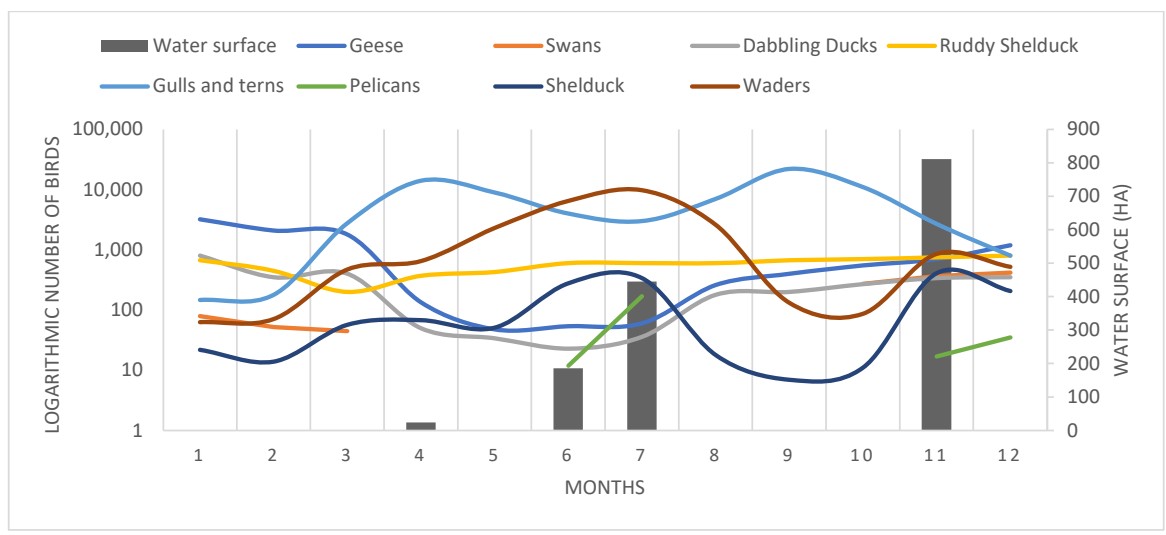

**Figure 7.** Average number of birds per the months in 2020.

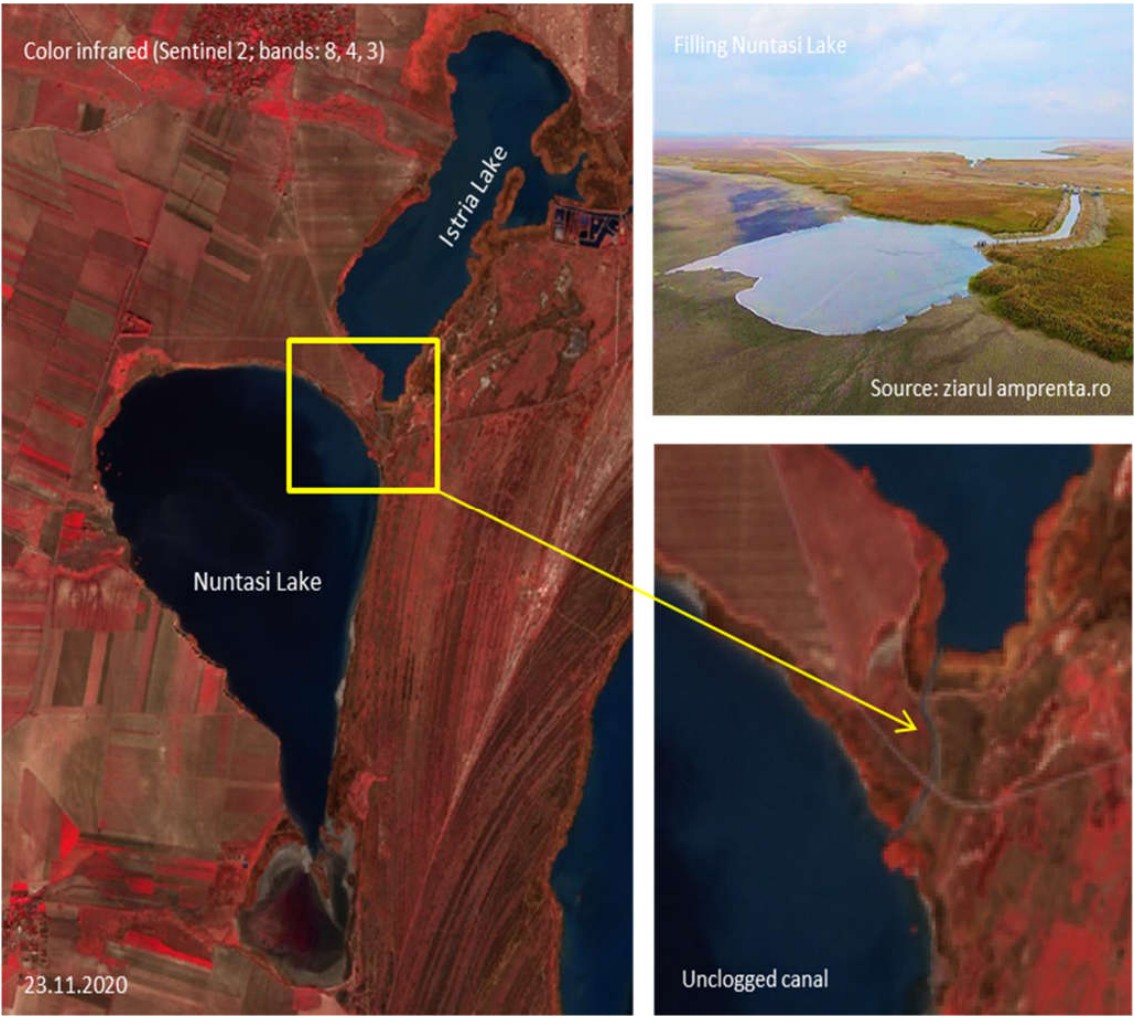

**Figure 8.** Nuntași and Tuzla Lakes after refilling process.

Although our initial analyses of 2013 and 2020 led us to think that in dry years the number of birds in our study area decreased, a simple analysis of the average number of birds over a period of 10 years (2010–2020) revealed that, in the years when the aquatic surface of the lakes decreased, the size of the flocks of some groups of birds increased compared to the years when the WSs of the lakes were not affected by drought (Figure 9). Even in this situation, pelicans were the most affected and only small numbers of them were identified in 2013 and 2020 (Figure 9).

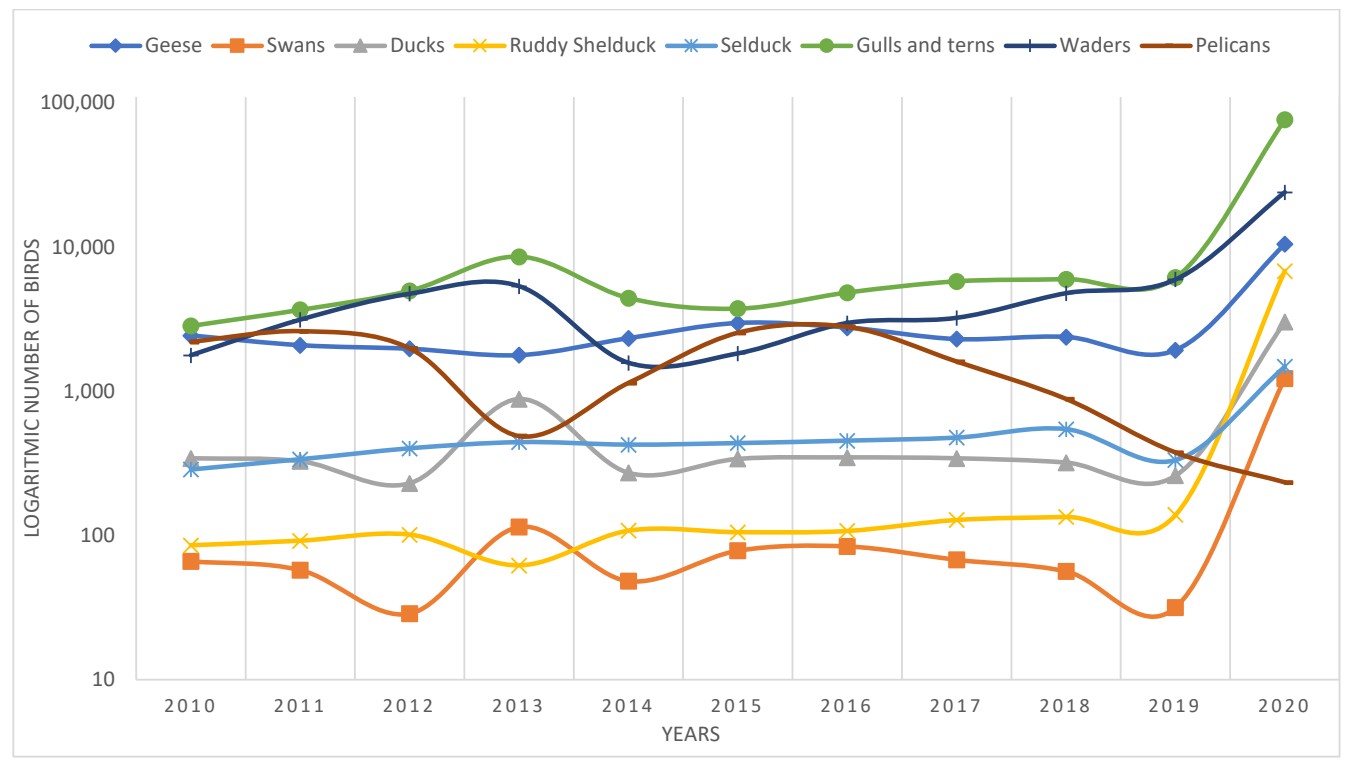

**Figure 9.** Average number of bird assemblages from 2010–2020.

Furthermore, a Kruskal–Wallis test showed that between 2020 and some of the other years (2010, 2014 and 2019) there is a statistically significant difference ($p < 0.005$) concerning the bird assemblage (Table 3) with regard to the WS [47].

**Table 3.** Kruskal–Wallis test regarding birds assemblages among years 2010–2020.

| *p*-Values: | 2010 | 2011 | 2012 | 2013 | 2014 | 2015 | 2016 | 2017 | 2018 | 2019 | 2020 |
|---|---|---|---|---|---|---|---|---|---|---|---|
| 2010 | 1 | 0.807 | 0.822 | 0.611 | 0.969 | 0.725 | 0.551 | 0.653 | 0.674 | 0.883 | 0.031 |
| 2011 | 0.807 | 1 | 0.984 | 0.792 | 0.777 | 0.914 | 0.725 | 0.837 | 0.860 | 0.922 | 0.056 |
| 2012 | 0.822 | 0.984 | 1 | 0.777 | 0.792 | 0.899 | 0.710 | 0.822 | 0.845 | 0.938 | 0.054 |
| 2013 | 0.611 | 0.792 | 0.777 | 1 | 0.584 | 0.876 | 0.930 | 0.953 | 0.930 | 0.717 | 0.100 |
| 2014 | 0.969 | 0.777 | 0.792 | 0.584 | 1 | 0.695 | 0.525 | 0.625 | 0.646 | 0.853 | 0.028 |
| 2015 | 0.725 | 0.914 | 0.899 | 0.876 | 0.695 | 1 | 0.807 | 0.922 | 0.945 | 0.837 | 0.072 |
| 2016 | 0.551 | 0.725 | 0.710 | 0.930 | 0.525 | 0.807 | 1 | 0.883 | 0.860 | 0.653 | 0.120 |
| 2017 | 0.653 | 0.837 | 0.822 | 0.953 | 0.625 | 0.922 | 0.883 | 1 | 0.977 | 0.762 | 0.089 |
| 2018 | 0.674 | 0.860 | 0.845 | 0.930 | 0.646 | 0.945 | 0.860 | 0.977 | 1 | 0.784 | 0.083 |
| 2019 | 0.883 | 0.922 | 0.938 | 0.717 | 0.853 | 0.837 | 0.653 | 0.762 | 0.784 | 1 | 0.045 |
| 2020 | 0.031 | 0.056 | 0.054 | 0.100 | 0.028 | 0.072 | 0.120 | 0.089 | 0.083 | 0.045 | 1 |

For 2013 and 2020, the Poisson regression showed that all of the bird assemblages were generally influenced by the meteorological data (the frequency of precipitation, temperature mean, maximum temperature and minimum temperature) and the WS variations (Table 3). Moreover, the changes in the weather parameters and fluctuation of the WS influenced the birds' presence on the Nuntași–Tuzla lakes. This finding achieved $p < 0.05$, which means

that there is a significant influence of the intercept among the meteorological variables on bird assemblages on the one hand and the WS on the birds' presence on other hand.

## 5. Discussion

Although the use of satellite imagery to track the evolution of water bodies is an already widely used method, either we focus on large areas over long periods of time [54,55] or we are talking about smaller areas [56,57], as in our case. However, we must recognize that the involvement of satellite remote sensing in the dynamics of avifauna as correlated with field data and meteorological data is a category of complex studies that require numerous resources, both in terms of the staff involved and the financing of the undertaking [58,59]. Regarding what we just stated, it could be considered an advantage to approach such a subject with the help of satellite images because, in the absence of data from specific institutions, they allow us to analyze a phenomenon that occurred in the past, without having to involve a large number of staff or a large financial effort. On the other hand, the frequent monitoring of the birds in an area such as the studied one involves both a large number of people and a considerable budget, which would otherwise be limiting factors. However, field monitoring also reveals other observations that can be particularly important in certain circumstances.

Regarding the bird assemblages in the Nuntași–Tuzla area, their presence is determined by their feeding and roosting possibilities, which depend on the meteorological factors or the WS area [60]. In this sense, waterfowl are influenced by weather factors of course, but it is important to emphasize that each group of birds reacts differently depending on their ecology. Climate fluctuations seem to be manifested by characteristics that tend to extremes, with a higher frequency of heatwaves [61], which is certainly reflected in the abundance and distribution of the groups of birds that were targeted in this study.

Moreover, knowing the dynamic pattern of the numerical changes and dispersals of the birds leads to an understanding of the time periods and weather conditions that can induce changes in the presence of the birds and, consequently, can establish ways of satellite monitoring combined with field monitoring and active conservation measures in order to keep a high diversity index. A high value of the diversity index (which captures the number of ecological groups and number of birds) can ensure the premises for long-term biodiversity conservation in accordance with European strategies [62], but also for the purpose of designating the Danube Delta Biosphere Reserve [63].

The fact that, in the case study, all of the meteorological factors generally influenced the presence of the birds induces the idea that the birds in the area are strictly dependent on the atmospheric conditions. However, the birds may be versatile and may find food and roost in adjacent areas regardless of the climatic parameters. On a large scale, previous studies [64] have stated that, in the context of climate change, long-distance migratory bird species will suffer less than sedentary species, precisely because of their mobility. It should also be noted that sedentary bird species are much more numerous than migratory ones, which highlights the vulnerability of birds to climate change [65,66], but also the need for case studies in smaller areas.

The high mobility of the birds also induces a certain independence of them from the WS area in the Nuntași and Tuzla lakes, considering that large lakes are found nearby that are also part of the Danube Delta Biosphere Reserve. On the other hand, the Nuntași and Tuzla lakes are prone to strong fluctuations in their water levels and this is a major limiting factor for most submerged plants as shallow sites may dry out. A temporary reduction of the water level is frequently used to control submerged vegetation in reservoirs [67]. Obviously, water level fluctuations are particularly problematic for the vegetation of these lakes, where marginal and shallow water-covered places are the only suitable places for growing vegetation. If the water level is kept relatively constant, we may expect that submerged vegetation will survive in the previously mentioned locations. The lack/disappearance of submerged vegetation leads to an increase in the availability of nutrients in the water and to the intensification of floating plant layers. Severe reduction of the WS or even the

complete drying of the lakes, like in the 2020 case, results in the loss of fish and most plant species. Also, the increased frequency of droughts is associated with significant decreases in invertebrate diversity [68]. Consequently, the number/presence of herbivorous and piscivorous bird species are affected in the short term by the reduction of the aquatic surface or a complete dry out in the extremely dry years. In contrast, the bird species that feed on aquatic and benthic fauna are favored by these temporarily larger feeding habitats (the presence of mudflats) and higher prey density (a greater concentration of aquatic fauna).

On the other hand, if the lake has a lot of water, it will be an excellent roost place because it is close to the feeding areas that are represented by the excellent cereal fields for geese and winter swans.

Moreover, swans and pelicans are the most sensitive and are strictly dependent on water, as the results show. In case of a lack of water, swans and pelicans leave the area. The other groups can use the areas in the middle of the lakes without water or with very little water as a place of rest, considering that they can spot terrestrial predators from a distance. In general, the reduction of the water surface induces a change in the habitats and negatively affects the bird populations [69,70], but not in this particular case. The number of birds can decline due to the lack of adequate management measures over a longer period of time [71,72].

The lack of water in 2020 lasted for months, until the application of some hydrological management measures. This also indicates the need for rapid and immediate intervention for the better conservation of wetland areas in order to ensure the living conditions of waterfowl [70].

Under the conditions of water restriction, a very rich foraging resource area for many bird species (waders, shelducks, ducks, gulls and terns) is created as the lake dries, which explains the birds' large numbers in the case of the droughts in the studied years. At the same time, seagulls and terns take advantage of the small water surface area because their food is concentrated in a small volume of water. So, given the fact that the restriction of the water surface or even the drying up in one season of the studied lakes causes an increase in the number of birds, it makes us wonder if the same would happen if the lakes were dried up for a longer period of time. Would the birds leave the area due to a lack of food in the mudflats or would they continue to use it, but only for resting? How birds behave or respond to restrictive conditions may be a future direction of scientific investigation based on this idea. Hence the need for the water level to be controlled in the sense of reducing the water level when it is too high to be favorable for bird species [73]. Even if the water luster is missing but there is enough wet mud, it is an excellent substrate for the development of the many invertebrate species that determine the presence of large numbers of birds. At the same time, the partial or complete drying of the lake must be avoided for long periods of time, in order to reduce the disappearance of the birds from this area and their dispersion over areas that are not so favorable, given the unique characteristics of the lakes complex.

As the frequency of droughts increases, there is an increasing impact on wetlands and, consequently, for birds, as is shown in the results. Thus, we find it necessary to analyze satellite images in conjunction with the monitoring of bird groups so that, through proper management, the Nuntași–Tuzla area remains a permanent hotspot for birds in one of the most valuable wetlands in the world, the Danube Delta Biosphere Reserve.

## 6. Conclusions

The present study reveals the characteristics of the current context, a context that is dominated by both climate change and habitat change, either as a result of a natural course or as a result of anthropogenic influence. In order to keep up with the dynamics of these changes, the inclusion of satellite images and field data (climate factors and biodiversity data) helps a lot in following the phenomenon and in observing the repercussions.

Our study outlines the suitability of long-term waterbird observations as an efficient tool to validate the qualitative and quantitative hydrological and biological transformations of wetlands, eventually supporting studies that are aimed at investigating climate change

scenarios while creating synergies, as much as possible, between climatic and biological data. Corroborating ornithological and hydrological observations may eventually enhance the development of integrated monitoring programs that can offer more comprehensive information regarding climate change's impact on wetlands.

This paper considered the presentation of 2 scenarios, one from 2013 in which a partial drying was analyzed and the other one from 2020 in which the total drying of the lakes for a period longer than 2 months was analyzed. The fluctuations of the water surface in the Nuntași and Tuzla lakes show an impact both on the submerged vegetation and on the vegetation near the lakes and, in the scenario of a complete drought, the impact that extended to fish and crustaceans, the potential food sources for the analyzed bird groups. Swans were the most affected by a total drought, leaving the area, which proves their dependence on the presence of water. A similar situation was encountered in the case of pelicans. Conversely, the other groups of birds tended to use the center of the dry lake as a resting area, having the opportunity to observe predators from a distance.

Given similar scenarios in the future, this paper will represent a starting point for analyzing the impact of such phenomena on birds and it will be easier to follow their behavior and their habitat preferences.

Given the fact that the area near the Nuntași and Tuzla lakes is suitable for birds (to the north, west and south there is agricultural land for feeding and to the east there is the Istria and Sinoie lakes) and that the saltier water delays the freezing of the lakes, these factors make this location a biodiversity hotspot. It is the case studies from 2013 and 2020 that highlight the fragility of this ecosystem in the context of changing natural factors and, at the same time, the need to involve human society in habitat conservation.

**Author Contributions:** Conceptualization, V.J., P.I. and C.I.; methodology, V.J., P.I. and C.I.; software, V.J., P.I. and C.I.; validation: V.J. and A.D.; formal analysis, V.J., P.I. and C.I.; investigation, V.J. and A.D.; resources, P.I., A.D. and C.I.; data curation, V.J., P.I., A.D. and C.I.; writing—V.J., C.I. and A.D.; writing—review and editing, V.J., P.I. and A.D.; visualization, P.I. and V.J.; supervision, V.J.,C.I. and P.I. All authors contributed equally and have equal rights to this research paper. All authors have read and agreed to the published version of the manuscript.

**Funding:** This research received no external funding.

**Institutional Review Board Statement:** Not applicable.

**Informed Consent Statement:** Not applicable.

**Data Availability Statement:** The data presented in this study are available on request from the correspondence author.

**Acknowledgments:** This project received technical support from the Department of Geography, Faculty of Geography and Geology, Alexandru Ioan Cuza University of Iasi, Romania, which offered us full access to the remote sensing and GIS laboratories. Vasile Jitariu is supported by the Romanian Young Academy, which is funded by Stiftung Mercator and the Alexander von Humboldt Foundation for the period 2021–2022. Pavel Ichim is supported by the Romanian Young Academy, which is funded by Stiftung Mercator and the Alexander von Humboldt Foundation for the period 2020–2022. Acknowledgment is also given by C.I. to the infrastructure support from the Operational Program Competitiveness 2014–2020, Axis 1, under POC/448/1/1 research infrastructure projects for public R&D institutions/Sections F 2018, through the Research Center with Integrated Techniques for Atmospheric Aerosol Investigation in Romania (RECENT AIR) project, under grant agreement MySMIS no. 127324.

**Conflicts of Interest:** The authors declare no conflict of interest.

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
