# Peer review of "Severe Drought Monitoring by Remote Sensing Methods and Its Impact on Wetlands Birds Assemblages in Nuntași and Tuzla Lakes (Danube Delta Biosphere Reserve)"

_land, doi:10.3390/land11050672_

Round 1

Reviewer 1 Report

Main comment:

The paper is well structured, starting from the presentation of the challenges related o the avifauna study and management applications. The site chosen for analysis and validation was based on the existence of data collected. During the research, adequate and up-to-date literature was consulted. Conclusions are clear and valuable for further researchers in the field. The technology used in water attribute estimation is innovative and reasonably cost-efficient compared to actual field observations. This approach has broad application potential. Personally, I believe this paper focuses too much on methods and ignores environmental and ecological issues.

Specific remarks:

Below you will find several comments, questions, and suggestions. The methods employed in the study were quite proper, but sometimes the information needed to fully document the conducted research appears to be incomplete.

1. The research on aquatic and remote sensing aspects and birds assemblages is too few, and the main application fields or directions of this method are not introduced. At this point, at least one of the hypotheses and the key calculation result referring to the main goal of the work is needed.

2. This article does not give an adequate introduction to compressed sensing, the concepts of sparsity, observation (measurement) matrix, and subsampling are not clearly explained, and the basic processes of resampling and signal reconstruction is not explained clearly. The scope of the whole method is not clear. The paper cannot fully explain the essence of the remote sensing method for ecological application, so it is necessary to improve and add more words to explain the methodology.

3. Poisson regression between bird assemblages and meteorological data is not clearly introduced in this paper, and the combination point of the two methods can not be obtained from the context. Due to the small number of sampling points, it is recommended to conduct fold cross-validation for the drought phenomenon, set instead of accumulating the sampling points (divided into four subsets), which cannot be explained clearly. Considering the complexity of satellite imagery to track the evolution of water bodies it is absolutely not the easiest way to estimate the bird abundance protection and make decision plans. The methodology should be clearly described, the model must be calibrated and the analysis framework ought to be properly described.

4. In Figure 6, the parameters do not seem clear, the display is confusing, and it needs improvement. Overall, the introduction requires improvement. The purpose of this chapter is to focus on the original topic and verify whether the assumptions are valid in light of the overall scientific goal.

5. The sketch map of the first data set should be included. It is not necessary to explain the text introduction of specific coordinate information in the text but to put a map of the geographic area. The basic details about the hydrological feature in the text are lacking. It is necessary to have information on the hydrological conditions in order to make informed decisions about bird protection. The method needs to be completed.

6. The discussion session is not deep enough. This paper's conclusion should be discussed in conjunction with the essence of compression sensing and related research.

Constructive feedback:

In the Conclusions section, more needs to be said than just repeating the results. This includes a comprehensive assessment of the data. It would be helpful to compare afforestation levels across the lake. Water retention may then be quantified. The author should specify where the data came from, where the measurements were taken, and whether they covered the same time period as the data on meteorological conditions. How were they resolved, homogeneous, and did they meet the fundamental assumptions of the drought appraisal model?

My comments are both specific and general:

How was the study area determined? What led to the selection? What are the main reasons? Is there any similar location that could be compared?

Is there any more recent data available?

What could be the next steps of this work? How could be used such an approach in the future? Please discuss more this point in the Conclusion and discussion section. Is this a practical approach? If so, what are the benefits and limitations?

At least one directional hypothesis and a research hypothesis are required. The results of the model calibration and the qualitative parameters should also be included. Using remote sensing as the basis for modeling, the entire "Results" section should be prepared from the very beginning. In managing a fragile ecosystem, many pertinent items must be completed in the field of hydrological modeling. The addition of any of my suggestions would redefine the scope of the paper and in some cases would require a different modeling approach and more laboratory data. As a result, conclusions drawn from the study area cannot be applied globally.

Summary:

This paper is a scholarly work and can increase knowledge in this domain. The authors provide an interesting and original study, the content is relevant to Land. The manuscript is quite well written and related to existing literature. The abstract and keywords are meaningful. However, this manuscript can not be fully accepted and requires a few amendments. The work described herein from a quality and from high interest, it's in the spotlight of the current research approach in this domain.

Reviewer 2 Report

#1 L19-20: be more specific. what do you mean by inadequate management?

#2 L28-29: What is the broader implication of this study? What are the projected changes in the climate (precip./temperature/water-surface) in the study region? What can we infer from the results/conclusion of this study about the fate of birds in the regions in the future climate?

#3 Figure 1: I suggest authors also show the map together with the world or continent map.

#4 L118-119: mention why it’s difficult to get scenes (e.g., overcast, cloud cover condition).  This is a very short paragraph(only one sentence). I suggest developing more or merging with another paragraph.

#5 Figure 2: In addition to the seasonality of precipitation, time series of regional average precipitation/drought would be helpful to visualize what is happening to the climate/meteorology of the regions  

#6 L228-233: Put the pictures/photos of these birds (mentioned in Figure 5,7,9) in a single figure.

#7 Table 3: replace , with .

#8 L329 “as in our case”.. Does that mean a similar analysis is already done in [51 and 52]? Then what is the novelty of your study? Improve the sentence

#9 L422-428: As in the abstract conclude the study with broader implications of the findings.

Reviewer 3 Report

The authors presents an interesting study showing the relationships between environmental (and human-induced) drought with the presence of waterfowl. They use the combination of field data (bird censuses) with data coming from meteorological stations and satellite imagery. In a situation of climate emergency, this study is so pertinent in order to manage the available water on wetlands. 

Anyway, to become a paper for Land journal, authors should change and improve some aspects of the ms.

General Aspects:

  1. In introduction section I miss some sentences dealing with the importance of water for waterfowl, specially during migration stopover.
  2. The second section should be study area and after that methodology. First bird census, second meterological data, then satellite imagery and finally data analysis (instead of methodology).
  3. Biological data, that should be bird census should be more detailed. As it is now, it's not clear when the census were performed: weekly ? monthly ? a what hours ? when along the year ? I think it's important.
  4. Tables and figures: there are too much, for sure that some could be omitted (see specific aspects).
  5. Appendix A is not necessary.

Specific aspects:

  1. Figure 1 has no title and no source. I recommend to add a small map of Romania in the European context. indicating where is the Danube Delta.
  2. Figure 2 and table 3 are not necessary. 
  3. The last sentence of birds observation do not belong to this section, better at climatic background.
  4. Figure 4 is a little bit confused. ES is WS ? The, I keep the Landsat 5 image and erase the right column. The graph showing the evolution of water surface area should switch to the text and erase. 
  5. Figure 5. Due to you don't have data from all the months and there is no clear trend of birds, I recommend move it in a text format rather than a graph. 
  6. The same that I've indicated on point 4, keeping only the satellite image. 
  7. Figure 7. The same that point 5.
  8. Figure 9 is strange due to theoretically 2020 was a drought year bbut the number of birds is the higher.
  9. I can't understand table 3.
  10. Due to table 4 is completely significative, is not necessary to show.
  11. Lines 328 and 329, you use "we are talking about", is better to use some words more academic. 
  12. Line 337, meteorological fluctuations better that cilmatic fluctuations.

Reviewer 4 Report

Due to the editor's reminders and limited time, I read the manuscript. I think this study is interesting and can pique the interest of the reader. In terms of the structure, research method and conclusions, I believe that the paper has reached the required level of Land.

Round 2

Reviewer 1 Report

Thank you for providing your revised text. Based on your comments, I think this version should be recommended.

Reviewer 3 Report

Thank you for the steady responses. In my opinion could be accepted in present form.

Congratulations.